# A systematic exploration of bacterial form I rubisco maximal carboxylation rates

Benoit de Pins [ID] [1], Lior Greenspoon [ID] [1], Yinon M Bar-On [ID] [1,4], Melina Shamshoum [ID] [1], Roee Ben-Nissan[1], Eliya Milshtein [ID] [1], Dan Davidi[1,5], Itai Sharon [ID] [2], Oliver Mueller-Cajar [ID] [3], Elad Noor [ID] [1] & Ron Milo [ID] [1✉]

## Abstract

**Autotrophy is the basis for complex life on Earth. Central to this process is rubisco—the enzyme that catalyzes almost all carbon fixation on the planet. Yet, with only a small fraction of rubisco diversity kinetically characterized so far, the underlying biological factors driving the evolution of fast rubiscos in nature remain unclear. We conducted a high-throughput kinetic characterization of over 100 bacterial form I rubiscos, the most ubiquitous group of rubisco sequences in nature, to uncover the determinants of rubisco's carboxylation velocity. We show that the presence of a carboxysome $CO_2$ concentrating mechanism correlates with faster rubiscos with a median fivefold higher rate. In contrast to prior studies, we find that rubiscos originating from $\alpha$-cyanobacteria exhibit the highest carboxylation rates among form I enzymes ($\approx 10\ \mathrm{s}^{-1}$ median versus $<7\ \mathrm{s}^{-1}$ in other groups). Our study systematically reveals biological and environmental properties associated with kinetic variation across rubiscos from nature.**

**Keywords** Carbon Fixation; Carboxysome; Cyanobacteria; Photosynthesis; Rubisco
**Subject Categories** Biotechnology & Synthetic Biology; Plant Biology

## Introduction

Biological carbon fixation is the gateway for food production and energy storage in the living world. Over 99% of global carbon fixation is catalyzed by rubisco (Raven, 2009), probably the most abundant enzyme in the biosphere (Bar-On and Milo, 2019). Rubisco is mainly divided into four distinct forms (I, II, II/III, and III) and can be found in all domains of life, from plants to algae through autotrophic bacteria and archaea (Prywes et al, 2023). Within this diversity, form I is by far the most abundant of the four forms: it is used by all plants and cyanobacteria and is responsible for almost all $CO_2$ fixation in nature.

Paradoxically, while being the most abundant, rubisco is probably the slowest (i.e., low maximum carboxylation rate - $k_{cat,C}$) central metabolic enzyme (Flamholz et al, 2019; Davidi et al, 2020). A systematic sampling of rubisco's genetic diversity can help grasp the boundaries of its carboxylation rate.

Our group has recently developed an approach to systematically explore the carboxylation rate of natural rubiscos. We use computational methods to select representative rubiscos from the tremendous sequence diversity space. Gene synthesis is then used to generate expression constructs encoding many rubiscos for purification and kinetic characterization. In previous work, we showed the feasibility of this approach by exploring form II and II/III rubisco variants (Davidi et al, 2020). We found an uncharacterized rubisco that has a $k_{cat,C}$ higher than all previously-known rubiscos—demonstrating the potential of this approach to stretch the kinetic boundaries of this pivotal enzyme.

In this work, we expand our search to form I rubiscos, which represent $\approx 97\%$ of known sequences (Davidi et al, 2020). Moreover, their immense ecological and sequence diversity (Tabita, 1999; Badger and Bek, 2008; Witte et al, 2010), limited kinetic data, and higher carboxylation rates in comparison to plant rubiscos (Flamholz et al, 2019), make bacterial form I variants particularly interesting. By conducting a first-of-its-kind large-scale study of uncharacterized bacterial form I rubiscos, and leveraging available meta-data on their sequences, we find correlations between contextual factors (phototrophy, carboxysome association) and fast carboxylating rubiscos.

## Results

### Large-scale survey of bacterial form I rubisco

To map the natural diversity of rubisco sequences, we performed an exhaustive search for rubisco homologs across the major genomic and metagenomic public databases. A total of 4300 unique sequences were identified as bacterial form I rubiscos (see "Methods" for more details).

By clustering systematically and at progressively higher sequence identity thresholds across different rubisco subgroups

[1]Department of Plant and Environmental Sciences, Weizmann Institute of Science, Rehovot 76100, Israel. [2]Migal Galilee Research Institute, Kiryat Shmona 11016, Israel. [3]School of Biological Sciences, Nanyang Technological University, Singapore 637551, Singapore. [4]Present address: Division of Geological and Planetary Sciences, California Institute of Technology, Pasadena, CA 91125, USA. [5]Present address: Aleph, Tel Aviv-Yafo 6688210, Israel. ✉E-mail: ron.milo@weizmann.ac.il

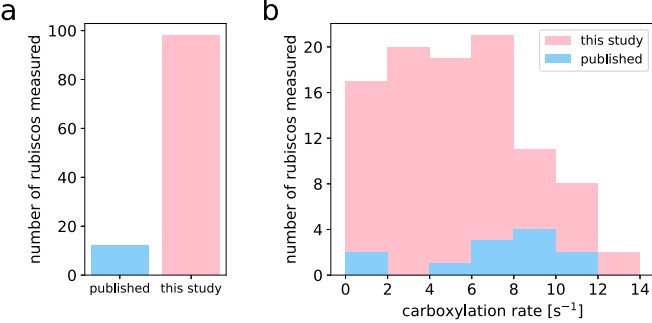

**Figure 1. Systematic exploration of the diversity of bacterial form I rubisco.**

(A) Number of bacterial form I rubisco variants with a carboxylation rate reported across the literature and in this study. (B) Histogram of the carboxylation rates measured in this study and across the literature. Source data are available online for this figure.

(as detailed in "Methods" and Fig. EV1), we selected 144 rubisco variants that represent the full spectrum of bacterial form I rubisco sequence diversity. In comparison, prior work using active site quantification, a method that allows for precise measurements of turnover rates while considering the enzyme's activation state, has so far characterized only 12 bacterial form I rubiscos (Fig. 1A; Dataset EV1).

In contrast to form II, II/III, and III rubiscos which are composed of a $\approx 50$ kDa large (rbcL) subunit organized as a homodimer, form I rubiscos comprise an additional $\approx 15$ kDa small subunit (rbcS) in an $L_8S_8$ stoichiometry. Due to this complex oligomeric structure, recombinant expression of form I rubiscos is challenging (Aigner et al, 2017). To improve correct folding in *Escherichia coli*, we coexpressed rubiscos with the chaperone GroEL-GroES, which is known to help reconstitution of bacterial form I rubiscos (Liu et al, 2010; Goloubinoff et al, 1989; Greene et al, 2007). In the case of variants originating from β-cyanobacteria, we coexpressed rbcL and rbcS together with their cognate chaperone rbcX whenever that gene was present in the operon. In addition, because only about a third of these variants were soluble initially, we screened different homologs of the rubisco accumulation factor 1 (Raf1), a chaperone that mediates the assembly of β-cyanobacterial rubiscos (Huang et al, 2020; Xia et al, 2020; Hauser et al, 2015; Kolesinski et al, 2014). We identified one homolog, from the bacteria *Euhalothece natronophila*, whose co-expression nearly doubled the number of solubly-expressed β-cyanobacterial rubiscos (Appendix Fig. S1A). We found that the average rate of β-cyanobacterial rubiscos was not changed by adding these newly soluble rubiscos (Appendix Fig. S1B).

The carboxylation rates of each expressed rubisco were determined using a modified version of the spectroscopic coupled assay reported at Davidi et al, 2020. Here, we directly assayed the crude cell lysates without purifying the enzyme in order to best preserve the quaternary structure of form I rubiscos. The method allows determining the specific carboxylation rate even without purification thanks to the use of the rubisco inhibitor CABP (see "Methods" and Appendix Note S1). Since the assay uses high $CO_2$ levels (4%), which is above the $K_M$ for most rubisco variants (Savir et al, 2010), measured rates are predicted to approach the $k_{cat,C}$

values. In Fig. EV2, we compare the rates of five rubisco variants with previously published values to the measurement in our lab, showing a similar ranking of carboxylation rates despite the different temperatures and assay methods.

Out of 144 rubisco variants tested, 112 were successfully expressed and soluble. Of which, 98 exhibited significant catalytic activity (which we define as >0.5 reactions per active site per second) (Dataset EV2). The median rate among active form I rubiscos was 5.4 $s^{-1}$, similar to the median $k_{cat,C}$ of plant rubisco literature values (4.7 $s^{-1}$ when corrected to 30 °C by assuming a Q10 value of 2.2 (Cen and Sage, 2005)). The fastest rate measured in this study was 13 $s^{-1}$ (Fig. 1B).

Altogether, our measurements achieve a nearly tenfold increase in the number of bacterial form I rubiscos with measured carboxylation rates. As described below, our results allow for the discovery of features associated with fast-form I bacterial rubiscos.

## Form I rubiscos from phototrophic bacteria are faster carboxylases than those from chemotrophs

Form I rubisco-expressing bacteria convert oxidized inorganic carbon ($CO_2$) into reduced organic compounds. To fuel the energy-intensive reactions involved in this process, they can draw upon two different energy sources: light (phototrophy) or chemical reactions (chemotrophy). Through a literature survey (see "Methods" and Dataset EV3), we collected available information on the bacteria expressing the studied rubiscos and classified them into phototrophs and chemotrophs. We tested whether one of these trophic modes was linked to higher carboxylation rates. To refrain from selection biases, we aimed to uniformly sample rubiscos from both classes as detailed in "Methods". We find that rubiscos originating from phototrophs have a carboxylation rate of 6.5 $s^{-1}$ [4.4–7.9 $s^{-1}$] (median and interquartile range), about three times faster than rubiscos derived from chemotrophs (2.1 $s^{-1}$ [1.4–4.0 $s^{-1}$]) (Fig. 2A, Mann–Whitney *U* test, $P < 0.01$). The same pattern was observed when taking together all rates measured in this study, without care of uniformly covering both groups (Appendix Fig. S2A).

## Carboxysome-associated form I rubiscos are significantly faster

A biological parameter that could have influenced rubisco evolution is the presence of a carboxysome-based $CO_2$ concentrating mechanism (CCM). This cellular mechanism combines the active transport of inorganic carbon into the cell and the colocalization of carbonic anhydrase and rubisco inside subcellular proteinaceous microcompartments called carboxysomes, locally increasing $CO_2$ concentration around rubisco (Flamholz and Shih, 2020). High $CO_2$ levels inhibit oxygenation by competitive inhibition, which can permit the use of less $CO_2$-affine but faster rubisco variants, following the observation of a kinetic tradeoff between these two parameters (Jordan and Ogren, 1981; Iñiguez et al, 2020).

To compare carboxysomal and non-carboxysomal form I rubiscos, we uniformly sampled rubiscos from each class and compared their measured carboxylation kinetics (see "Methods"). We found that with a median catalytic rate of 7.4 $s^{-1}$ [5.2–8.3 $s^{-1}$], carboxysome-associated rubiscos are more than five times faster than their non-carboxysomal counterparts (1.3 $s^{-1}$ [1.1–2.1 $s^{-1}$])

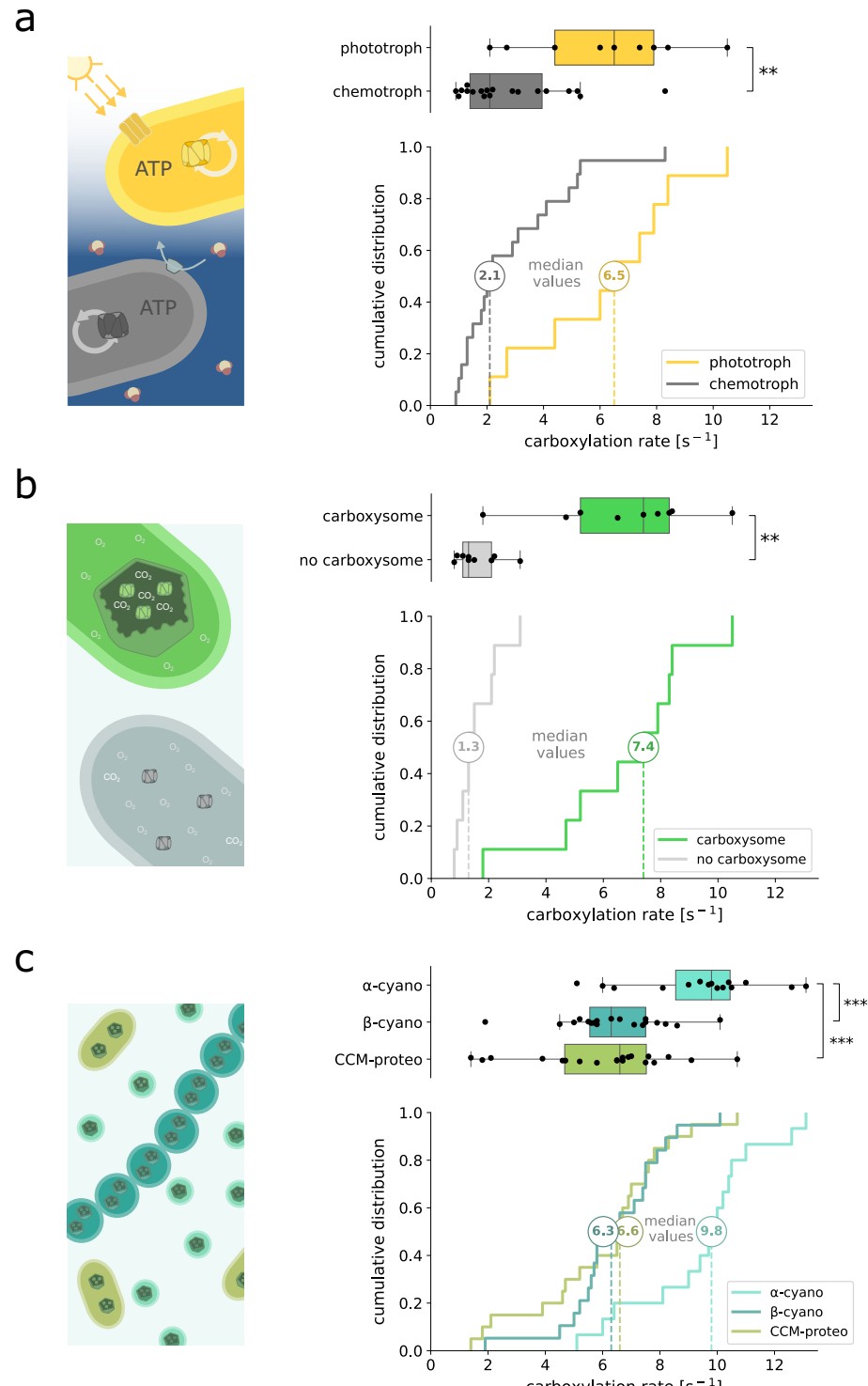

**Figure 2. Large-scale analysis of the biological parameters associated with fast carboxylating rubiscos.**

(A–C) Box and cumulative distribution plots of rubisco carboxylation rates from different clusters: (A) chemo- and phototrophic bacterial rubiscos, (B) carboxysome-associated rubiscos and their counterparts, (C) α- and β-cyanobacterial, and carboxysome-associated proteobacterial rubiscos. To ensure unbiased study of every group, we selected $n = 19$ and 9 class-representative chemo- and phototroph-associated rubiscos, $n = 9$ and 9 class-representative carboxysome-associated and non-associated rubiscos, and $n = 15$, 19, and 20 class-representative α- and β- cyanobacterial, and carboxysome-associated proteobacterial rubiscos respectively (see "Methods"). Boxes represent the first and third quartiles, with the medians indicated by the central lines. Whiskers extend to the lowest and highest values within 1.5 times the interquartile range. Mann–Whitney U test (A, B) or Kruskal–Wallis followed by Dunn multiple comparison tests (C) were applied. **$P < 0.01$, ***$P < 0.001$. Legend abbreviations are as follows: α-cyano, α-cyanobacterial rubisco; β-cyano, β-cyanobacterial rubisco; CCM-proteo, carboxysome-associated proteobacterial rubisco. Source data are available online for this figure.

(Fig. 2B, Mann–Whitney *U* test, *P* < 0.01). We found consistent results when taking together all rates measured in this study, i.e., ignoring uniform coverage (Appendix Fig. S2B). An exemplary case within our dataset is the gammaproteobacteria *Hydrogenovibrio kuenenii*, which expresses two form I rubiscos. The carboxysomal rubisco has a rate of 8.3 s$^{-1}$, which is twice as fast as its non-carboxysomal counterpart (4.2 s$^{-1}$, Dataset EV2), thus exemplifying the impact of carboxysome association on rubisco carboxylation.

## Alpha-cyanobacteria express the fastest form I rubiscos across the tree of life

Carboxysomes are known to be expressed by all cyanobacteria and some chemotrophic proteobacteria (our current analysis also showed bioinformatically that it can be found in phototrophic proteobacteria such as the purple sulfur bacteria *Thiorhodococcus drewsii*). Cyanobacteria can be divided into two sub-clades, α- and β-cyanobacteria (Badger and Price, 2003; Whitehead et al, 2014). It has been posited that α- and β-cyanobacteria rubiscos had identical catalytic rates (Cabello-Yeves et al, 2022) or that β-cyanobacteria rubiscos are faster (Iñiguez et al, 2020; Nguyen et al, 2023), as they included the fastest form I rubisco characterized to date (from *Synechococcus elongatus* PCC 6301) (Savir et al, 2010). However, such statements were made based on scarce measurements, with only three $k_{cat,C}$ values currently available for both rubisco groups (Shih et al, 2016; Long et al, 2018; Sharwood and Long, 2021; Wilson et al, 2018; Aguiló-Nicolau et al, 2023). We now reevaluate this hypothesis using a wider and systematic kinetic sampling of these different rubisco subforms. We have class representative subsets of 15, 19, and 23 variants uniformly covering α-, β-cyanobacteria, and proteobacteria carboxysome-associated rubiscos diversity from our dataset (see "Methods").

We find that, in contrast to previous statements, α-cyanobacterial rubiscos show the highest carboxylation rates (9.8 s$^{-1}$ [8.6–10.5 s$^{-1}$]) among all bacterial form I rubiscos, ≈50% higher than β-cyanobacterial rubiscos and their proteobacterial counterparts (6.3 s$^{-1}$ [5.6–7.5 s$^{-1}$] and 6.6 s$^{-1}$ [4.7–7.5 s$^{-1}$], respectively, Fig. 2C, Kruskal–Wallis test followed by Dunn multiple comparison test, *P* < 0.001). We observed the same result when taking together all rates measured in this study, regardless of achieving uniform coverage across groups (Appendix Fig. S2C). Future work can validate this result with direct assays and tests on the impact of other $CO_2$ concentrations, different temperatures, etc.

We further analyzed the correlation between rubisco carboxylation rate and various biological and environmental parameters such as bacterial environmental source, rubisco subtype, bacterial halotolerance, pH, oxygen sensitivity, or optimal growth temperature. As presented in Appendix Figs. S3–8, these showed much weaker or no correlations. For example, while pH is known to be crucial for carboxysome efficiency (Mangan et al, 2016; Long et al, 2021), the optimal growth pH of a bacterium does not show any correlation with its rubisco carboxylation rate (Appendix Fig. S6), likely as it does not directly affect the tightly controlled intracellular pH. In addition, the slightly lower carboxylation rate of rubiscos originating from thermophilic bacteria and isolated from hot environments (Appendix Figs. S3 and S8) aligns with expectations, considering that these rubiscos naturally work at higher temperatures than in our in vitro assay (30 °C). However, this is unlikely to explain the observed trends as the main results of this study are not

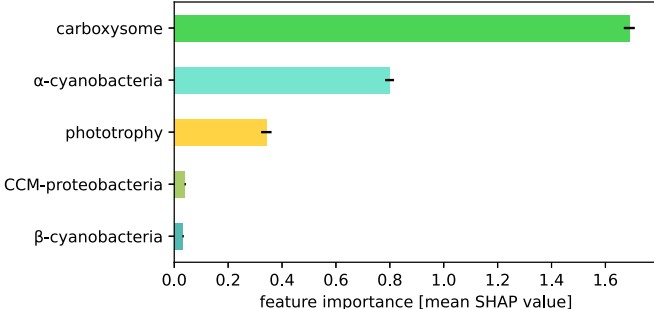

**Figure 3. The presence of a carboxysome is the primary factor influencing form I rubisco carboxylation rate.**

Feature importance was determined using absolute SHAP (Shapley additive explanations) values from a random forest regressor model. The model assessed the rubisco carboxylation rate based on bacterial trophic mode, carboxysome-association, and belonging to specific carboxysome-expressing bacterial group: alpha-cyanobacteria, beta-cyanobacteria, or carboxysome-associated proteobacteria (CCM-proteobacteria). Error bars are the standard deviations across 100 different train-test splits.

affected by the removal of these rubiscos (Appendix Fig. S9). Investigating the temperature response of rubisco carboxylation rate in further work could shed light on the importance of this parameter, especially among thermophilic or psychrophilic associated enzymes.

We eventually generated the structures of all 98 active rubiscos from this study. We did not detect striking differences between fast and slow rubiscos. A median RMSD value of 2.7 Å was found between the structures (rubisco large and small subunits together), and 1.3 Å for the active site alone (Appendix Fig. S10). These two values are close to the resolution of AlphaFold predicted structures (Jumper et al, 2021; Terwilliger et al, 2024), which would make it challenging to pinpoint any clear structural differences using these models.

To evaluate possible dependencies among the features showing correlations, we performed a joint analysis to see the contribution of each feature while accounting for the other features. In order to achieve this, we trained a random forest regressor model using our dataset to predict rubisco's carboxylation rate based on the main factors explored in this work (see "Methods" and Fig. EV3). The derived Shapley additive explanations (SHAP) values (Lundberg and Lee, 2017; Lundberg et al, 2020) quantify the influence of each feature on the predicted rate (Fig. 3). Among all the features considered in this study, carboxysome association is by far the most important one for determining the carboxylation rate of form I rubisco. This is followed by belonging to alpha-cyanobacteria. The influence of phototrophy, while present, is only marginal when correcting for the presence or absence of a carboxysome.

## Discussion

Rubisco is among the enzymes that have helped shape Earth's biosphere and geosphere the most. Its kinetic parameters result from billions of years of evolution, following (and causing) changes in the atmosphere and climate. We present here a systematic large-

scale survey of ≈140 rubisco variants covering the genetic diversity of bacterial form I rubiscos.

We note that the present study is limited by the risk of underestimating some rates if, for instance, enzymes are partially denatured in the expression conditions used. Yet, a denatured enzyme probably could not interact with RuBP or CABP, and more generally, we have no reason to presume a priori a systematic bias affecting some of the studied groups over others.

We find that carboxysome-associated rubiscos are, on average, more than five times faster than counterparts which are not associated with a carboxysome. Moreover, among the main parameters examined in this study, it exhibits the strongest association with the occurrence of fast carboxylating rubiscos. Carboxysomes likely evolved during the Proterozoic eon—in the context of the continuous decrease of carbon dioxide in Earth's atmosphere (Flamholz and Shih, 2020)—to maintain carboxylase activity in this changing atmosphere. One evolutionary strategy is the emergence of $CO_2$ concentrating mechanisms (CCMs), including carboxysomes, that locally concentrate $CO_2$ around rubisco and therefore maintain local gas concentrations favorable to carboxylation. Another strategy consists of the evolution of rubisco towards stronger affinity for $CO_2$ (Tortell, 2000). It has long been postulated that rubisco is a constrained enzyme, limited by catalytic tradeoffs, notably between its carboxylation rate and affinity for $CO_2$ (illustrated by the positive correlation between $k_{cat,C}$ and $K_{M,CO_2}$) (Jordan and Ogren, 1981; Iñiguez et al, 2020). Rubiscos that evolved in the context of a CCM might have faced less selective pressure towards stronger $CO_2$ affinity and have been postulated to present higher carboxylation rates as observed in $C_4$ versus $C_3$ plants (Seemann et al, 1984). Our findings with carboxysomal rubiscos support this conjecture using the most comprehensive comparative study of rubisco catalytic speed presented to date.

Eventually, among cyanobacteria, α-cyanobacterial rubiscos were found to be 50% faster than their β-cyanobacterial counterparts. One possible difference between α- and β-cyanobacteria could be their cell size. α-cyanobacteria are known to encompass many members of the so-called picocyanobacteria, the smallest cyanobacteria on Earth (Jasser and Callieri, 2017; Cabello-Yeves et al, 2022). We collected from the literature the size dimensions of cyanobacteria, when available. With a median cell volume of $0.5\,\mu m^3$ $(0.3-1.0\,\mu m^3)$, α-cyanobacteria are more than 25 times smaller in volume than their β counterparts $(13.5\,\mu m^3$ $[3.9-25\,\mu m^3])$ (Appendix Fig. S11, Mann–Whitney $U$ test, $P < 0.0001$). Smaller cells offer a higher surface-to-volume ratio and increased exchange with the medium (Young Kevin D, 2006), which could support a higher supply of nutrients, and contribute to the evolution of faster rubiscos in these specific cyanobacteria (see Appendix Note S2 for more details). We also note that α-carboxysomes are smaller than β-carboxysomes (Rae et al, 2013). The concentration of $CO_2$ molecules at rubisco active site may therefore be even higher in α-cyanobacteria.

By centering our analysis on the carboxylation rate, this pipeline systematically shows the particularity of carboxysome-associated rubiscos which are characterized by a poor affinity to $CO_2$ (Badger et al, 1998; Falkowski and Raven, 2007) alongside a relatively high carboxylation rate (our data). This likely reflects the aforementioned catalytic tradeoff, suggesting that higher local concentrations of $CO_2$ within CCMs probably allowed rubiscos to evolve towards

higher $k_{cat}$ and $K_M$. High-throughput measurements of other kinetic parameters beyond what was achieved here, such as the $K_M$ for both gasses or the oxygenation rate, would be valuable. It could provide values of the carboxylation efficiency, or even the enzyme specificity, which would enrich our understanding of this enzyme and of its adaptation to the atmospheric composition over geological timescales.

This study provides a systematic exploration of bacterial form I rubisco maximal rates and its relationship with various contextual factors that could have shaped the evolution of this most abundant enzyme on Earth. It holds potential for future metabolic and ecological studies about specific bacterial species—for instance, among cyanobacteria for which 40 rubiscos have been characterized here. By enriching our knowledge on carboxylation rates and their connection to environmental factors, it can also contribute to more accurately modeling global carbon fluxes. In addition, this dataset of rubisco sequences and their associated rates can facilitate linking sequence motifs to catalytic function. Ultimately, it can improve our understanding, and possible harnessing, of bacterial CCMs for the potential development of plant-based carbon capture strategies, the increase of agricultural yields and the support of sustainable food production in the face of a changing climate.

# Methods

## Rubisco sequence collection

Rubisco large subunit sequences were collected from (i) the NCBI's nr database (Benson et al, 2013) downloaded in December 2020 and searched following the method described in Davidi et al, 2020; (ii) in-house assemblies of the 244 samples from the Tara Oceans expedition (Sunagawa et al, 2015); (iii) assemblies and rubisco sequences published by Wrighton et al, 2012; Brown et al, 2015; Anantharaman et al, 2016; Tabita et al, 2008b; Banda et al, 2020. Sequences outside the length range of 300–700 amino acids were removed. The remaining sequences were then clustered at an 80% sequence identity threshold using USEARCH (Edgar, 2010). Cluster representatives were aligned using MAFFT (v7.475, default parameters) (Katoh and Standley, 2013), and columns with more than 95% gaps were removed using trimAl (v1.4.rev15, -gt 0.05) (Capella-Gutiérrez et al, 2009). A phylogenetic tree was constructed using FastTree (v2.1.10, default parameters) (Price et al, 2010). To identify the different rubisco forms, we relied on annotated sequences from NCBI, Tabita et al, 2008a, and Banda et al, 2020. This process resulted in a total of 72,395 sequences, including 56,161 form I rubiscos, and most notably for this study, 4302 non-eukaryotic form I rubiscos. The latter were further re-clustered at 90% identity using USEARCH algorithm (custom Python script; see below), and a phylogenetic tree was constructed using RAxML (Stamatakis, 2014).

## Rubisco variants selection for characterization

Form I rubiscos are divided into five separate groups (Badger and Bek, 2008; Tabita et al, 2008b; Spreitzer and Salvucci, 2002; Park et al, 2009; Grostern and Alvarez-Cohen, 2013) (see Appendix Fig. S4A): forms IA and IB (forming the "green" type, found in cyanobacteria and some proteobacteria); forms ICD and IE (the

"red" type, found in proteobacteria and some Terrabacteria respectively); and the recently discovered form I "Anaero" (found in bacteria related to anaerobic, thermophilic Chloroflexaeota and Firmicutes) (Schulz et al, 2022). To comprehensively sample the sequence space of form I rubisco diversity, an iterative clustering approach of the large subunit gene was employed using USEARCH algorithm. The resulting representative sequences from each cluster were selected for characterization. Thresholds were chosen in line with the number of variants we could afford to synthesize and measure in the span of this study. Initially, 32 rubiscos were chosen to cover the entire diversity of form I rubisco at a threshold of 75% identity. Subsequently, further clustering was performed on smaller groups of rubiscos of particular interest, with increasing threshold percentages. Throughout the study, this successively included 38 rubiscos representing the diversity of form IA and B rubisco at 85% identity, 13 rubiscos representing the diversity of cyanobacterial rubisco at 88% identity, 29 rubiscos representing the diversity of IB rubisco at 91% identity, 19 rubiscos representing the diversity of cyanobacterial IA rubisco at 97.5% identity, and 23 rubiscos representing the diversity of proteobacteria carboxysome-associated rubisco at 90% identity (see Fig. EV1). These representative sequences could sometimes overlap, resulting in a total of 129 different rubisco sequences tested in this study. In addition, 15 rubiscos were arbitrarily selected for setting-up the experimental pipeline. In total, 144 different rubisco variants were selected for characterization in this study.

## Gene synthesis

For each chosen rubisco variant, the complete rubisco operon, encompassing rubisco large and small subunit genes, as well as the chaperone *rbcX* gene for IB rubiscos, was retrieved. The operons were then codon-optimized for expression in *E. coli* (Twist Codon Optimization tool) and synthesized by Twist Bioscience. Following synthesis, these operons were cloned into a pET-29b(+) over-expression vector (NdeI_XhoI insertion sites). Validation of gene synthesis and cloning was conducted through next-generation sequencing as part of the Twist bioscience service.

## High-throughput rubisco expression

Chemocompetent BL21(DE3) cells, previously transformed with a pESL plasmid coding for the chaperone GroEL-GroES (Goloubin-off et al, 1989), were transformed with the rubisco library and incubated at 37 °C, 250 rpm in 8 ml of LB media supplemented with 30 μg/ml chloramphenicol and 50 μg/ml kanamycin. Growth was performed in 24-deep-well plates. When cells reached an $OD_{600}$ of 0.6, GroEL-GroES expression was induced by adding arabinose (0.2% final) and incubating at 23 °C for 45 min. Rubisco expression was then induced by adding 0.2 mM IPTG (isopropyl β-d-thiogalactoside) and incubating at 23 °C for 21 h. For protein extraction, cells were harvested by centrifugation (15 min; $4000 \times g$; 4 °C) and pellets were lysed with 70 μl BugBuster® master mix (Millipore) for 25 min at room temperature. Crude extracts were then centrifuged for 30 min at $4000 \times g$, 4 °C to remove the insoluble fraction. For quality control of each sample, 0.2 μl of the crude extracts and 2 μl of the soluble fractions (i.e., 20–40 ng of proteins) were run on an SDS–PAGE gel (Appendix Fig. S12).

## Raf1-IB rubisco co-expression

For IB rubiscos (originating from β-cyanobacteria), a pilot study was performed to find an homolog of the chaperone Rubisco accumulation factor 1 (Raf1), that could help solubly-express these rubisco variants. Three different Raf1 were first tested with their cognate rubisco, from 3 β-cyanobacteria (namely, *Trichormus variabilis*, *Pseudanabaena sp.* PCC 6802, and *Euhalothece natronophila*). Codon-optimized *raf1* genes were synthesized and cloned by Twist Bioscience in polycistron with their cognate rubisco genes into pET-29b(+) vectors. Proteins were expressed and tested in vitro, as described above, and Raf1 from *Euhalothece natronophila* was chosen, based on ability to solubilize its cognate rubisco. The gene was therefore cloned into a pESL plasmid, in the same operon as GroEL-ES genes, and transformed into chemo-competent BL21(DE3) cells. IB rubiscos were then transformed and expressed in these cells as described previously.

## High-throughput determination of rubisco carboxylation rates

To determine rubisco carboxylation rates, we performed kinetic assays directly from the soluble fraction of prepared lysates since purifying both large and small rubisco subunits together was not feasible in a high-throughput manner. Soluble fractions were incubated with 4% $CO_2$ and 0.4% $O_2$ for rubisco activation (15 min; 4 °C; plate shaker at 250 rpm). Following activation, rubisco carboxylase activities were tested as described in Davidi et al, 2020. Briefly, 10 μl of the activated rubisco sample was added to six aliquots of 80 μl of assay mix (a detailed list of all assay components and their sources is provided in Appendix Table S1) containing different concentrations of CABP (0, 0, 10, 20, 30, and 90 nM). The mix was incubated for 15 min at 30 °C in a plate reader (Infinite® 200 PRO; TECAN) connected to a gas control module (TECAN pro200) ensuring atmospheric conditions of 4% $CO_2$ and 0.4% $O_2$. Rubisco carboxylation activity was initiated by adding 10 μl RuBP to each sample (final concentration of 1 mM and a total volume of 100 μl). The carboxylation rate was determined through a coupled reaction (Lilley and Walker, 1974; Kubien et al, 2011). In brief, 3-phosphoglycerate, the product of ribulose 1,5-bisphosphate car-boxylation, was phosphorylated and subsequently reduced into glyceraldehyde 3-phosphate, involving NADH oxidation. The latter could be monitored through 340 nm absorbance for 15 min at 2-s intervals. Knowing the NADH extinction coefficient at 340 nm ($\varepsilon_{340} = 6220$ $M^{-1}cm^{-1}$), and after measuring the optical path length (l = 0.26 cm) with an NADH calibration curve in our setting, we used Beer-Lambert law ($A_{340} = \varepsilon_{340} \cdot l \cdot c$) to measure the NADH concentration c. To convert from NADH to rubisco reactions per second, we assumed a stoichiometric ratio of 2:1 between NADH and carboxylation reactions. The active-site concentration was determined by fitting a linear regression model (custom Python script; see below) to the measured reaction rates as a function of the CABP inhibitor concentrations (see Appendix Note S1; Appendix Fig. S13 for more details). Due to the use of soluble fractions of total cell lysates, the initial concentration of rubisco could not be determined beforehand, which often led to saturation of the first assay with rubisco. To overcome this issue, we adjusted the concentration of lysates by dilution, to obtain a rubisco concentra-tion that enabled measurable inhibition by CABP, allowing for an

accurate quantification of the active-site concentration. We finally obtained the rate per active site by dividing the activity with no CABP by the concentration of active sites. As was done in Davidi et al, 2020, and since not all variants were tested on the same day, the form-II rubisco from R. rubrum, commonly used as a standard in rubisco kinetic assays, was consistently included as an internal reference in every measurement.

## Bacterial, ecological, and rubisco data collection

Data were collected on the bacteria expressing rubiscos studied in this work. A literature survey was performed to gather as much available information on each bacteria. In addition, sample data associated with these bacteria were collected on NCBI (BioSample database). Furthermore, the identification of carboxysome and non-carboxysome-associated rubiscos was carried out by systematically examining the presence of carboxysome genes following the small subunit gene of rubisco, *rbcS*. All collected information and references are presented in Dataset EV3.

## Rubisco variants selection for comparative analysis

For comparing rubisco carboxylation rates between groups without sampling bias, we aimed to select representative sets of variants uniformly covering the different groups studied. For the comparison of rubiscos originating from phototrophic versus chemotrophic organisms, we selected a set of 32 rubiscos, covering the entire diversity of form I rubiscos at a 75% similarity threshold, enriched with a set of 36 rubiscos, covering the entire diversity of form IA and B rubiscos at an 85% similarity threshold. Among these 68 variants, phototrophic rubiscos were distinguished from chemotrophic ones based on the trophic mode of their host bacteria reported in the literature. This resulted in 52 chemotropic and 15 phototrophic rubiscos, of which 19 and 9, respectively, were soluble and active in vitro.

For the comparison of carboxysome and non-carboxysome-associated rubiscos, we selected a set of 14 rubiscos covering the entire diversity of carboxysome-associated rubiscos at an 85% similarity threshold, and of 30 rubiscos covering the entire diversity of non-carboxysome-associated rubiscos at a 75% similarity threshold. 9 and 9 of them were soluble and active in vitro.

For the comparison of α- and β-cyanobacterial, and carboxysome-associated proteobacterial rubiscos, we selected 19 rubiscos covering the entire diversity of α-cyanobacteria rubiscos at a 97.5% similarity threshold, 29 rubiscos covering the entire diversity of β-cyanobacteria rubiscos at a 91% similarity threshold, and 23 rubiscos covering the entire diversity of carboxysome-associated proteobacterial rubiscos at a 90% similarity threshold. 15, 19, and 20 of them, respectively, were soluble and active in vitro.

## Modelling rubisco structures and structural analysis

The structures of the 98 active rubiscos (one large and one small subunit: LS) were modeled using CollabFold implementation (Mirdita et al, 2022) of AlphaFold2 (Jumper et al, 2021) with the following settings: msa_mode: mmseqs2_uniref_env; num_models: 5; num_recycles: 3; use_amber. The hexadecameric $L_8S_8$ form was then predicted using Pymol v2.5 (https://pymol.org/2). To do so, the target large and small subunits LS were loaded together with the hexadecameric structure of *Synechococcus elongatus* PCC 6301 rubisco (Protein Data Bank code 1RBLl), and successively copied and aligned to every pair of large and small subunits from this reference structure (custom python script). Using Pymol, we measured the RMSD of both the LS structure and the active site independently (custom Python scripts; see below).

## Random forest regressor model and feature importance analysis

A random forest regressor model was used to predict rubisco's carboxylation rate as a function of the main features showing correlation in the study: the trophic mode of the bacteria (photo or chemotrophy), the association of the rubisco with a carboxysome, and, among those harboring a carboxysome, those belonging to proteobacteria, α- or β-cyanobacteria. One hundred individual decision trees were trained with a maximum depth of 3 and a fixed random seed of 42. In each tree generation, the dataset was randomly split into training (75% of the data) and testing (25%) sets. For every generated tree, Shapley additive explanations (SHAP) values were computed for each estimator. SHAP values represent the average contribution of each feature to the difference between the model's prediction and the measured value. The SHAP values of each parameter were eventually averaged and plotted for feature importance comparison.

## Data availability

All the data supporting the findings of this study as well as the 98 modeled protein structures, and the codes used for generating our list of rubiscos and for analyzing the results is open source and can be found in the following link: https://gitlab.com/milo-lab-public/rubisco-F1.

The source data of this paper are collected in the following database record: biostudies:S-SCDT-10_1038-S44318-024-00119-z.

## Peer review information

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

## Acknowledgements

The authors thank Yoav Peleg, Ron Sender, Noam Prywes, Brian Ross, Dina Listov, Ralf Steuer, Avi Flamholz and David Savage for important conversations and productive feedback on this manuscript. The authors thank Michelle Gehring for additional information about unpublished data from their laboratory. This research was supported by the Mary and Tom Beck Canadian Center for Alternative Energy Research, Miel de Botton, the Schwartz Reisman Collaborative Science Program, and the Charles and Louise Gartner Professorial Chair.

## Author contributions

**Benoit de Pins**: Conceptualization; Data curation; Investigation; Methodology; Writing—original draft; Writing—review and editing. **Lior Greenspoon**: Investigation; Methodology. **Yinon M Bar-On**: Methodology. **Melina Shamshoum**: Methodology. **Roee Ben-Nissan**: Methodology. **Eliya Milshtein**: Methodology. **Dan Davidi**: Methodology. **Itai Sharon**: Data curation; Investigation. **Oliver Mueller-Cajar**: Conceptualization; Supervision; Writing—original draft; Writing—review and editing. **Elad Noor**: Conceptualization; Supervision; Methodology; Writing—original draft; Writing—review and editing. **Ron Milo**: Conceptualization; Supervision; Funding acquisition; Methodology; Writing—original draft; Project administration; Writing—review and editing.

Source data underlying figure panels in this paper may have individual authorship assigned. Where available, figure panel/source data authorship is listed in the following database record: biostudies:S-SCDT-10_1038-S44318-024-00119-z.

## Disclosure and competing interests statement

The authors declare no competing interests.

# Expanded View Figures

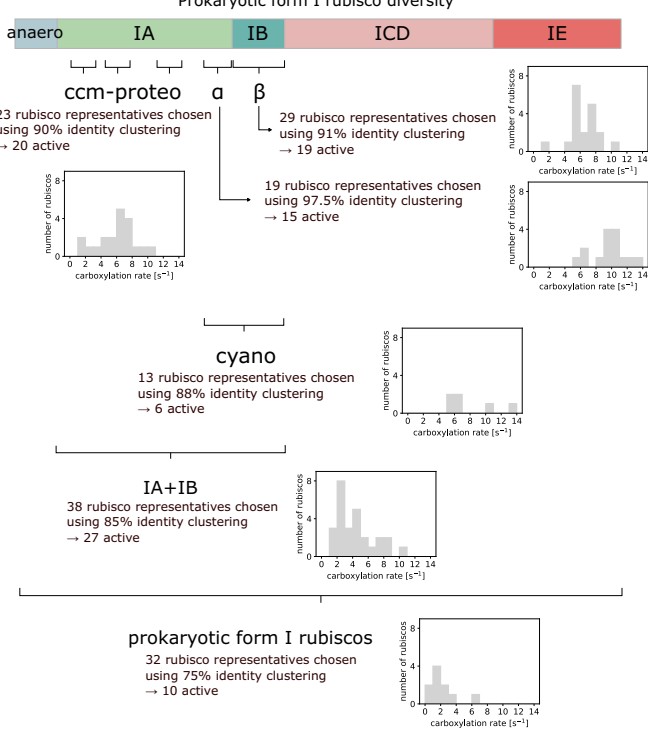

**Figure EV1. Sequential screening strategy used for the selection of rubisco variants for characterization.**

An iterative clustering approach was employed to screen the totality and specific subgroups of the form I rubisco family. The number of active rubiscos and measured carboxylation rates are indicated at each step.

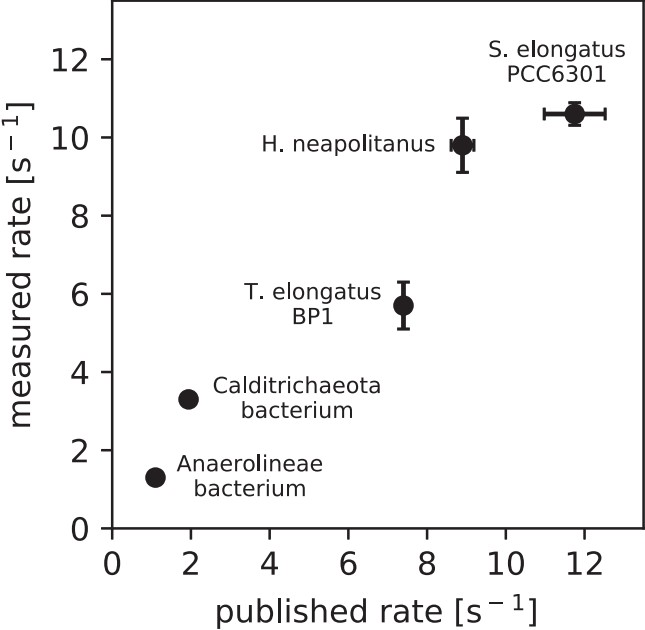

**Figure EV2. Carboxylation rates measured in this study have similar ranking as rates published in the literature in spite of different techniques and conditions.**

Values in both axes are not supposed to be equal as those measured in this study result from coupled assays, which tend to underestimate the rates compared to direct assays used in the literature, and on the other side they were performed at 30 °C, resulting in faster rates compared to literature measurements done at 25 °C.

## a

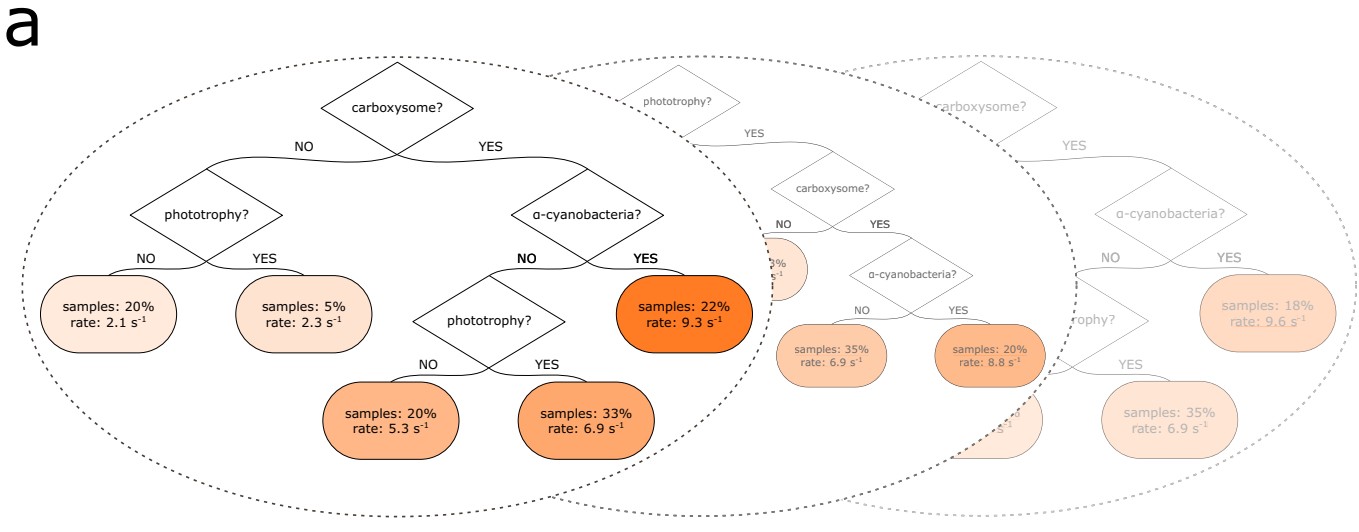

## b

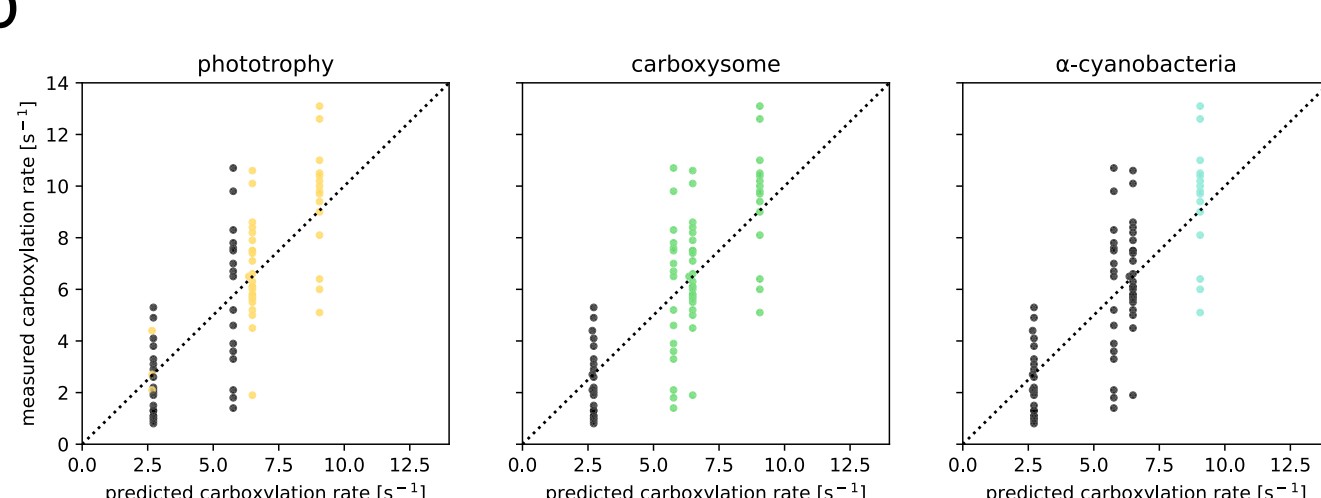

**Figure EV3. Random forests modeling of form I rubisco carboxylation rate as a function of the main parameters from this study.**

(A) Examples of decision trees out of the hundred trained in the random forests. (B) Measured against predicted carboxylation rates for the three most influential features according to the model. Each dot is color-coded based on the value of the respective feature (yellow/black: phototrophic/chemotrophic; green/black: carboxysome/non-carboxysome associated; cyan/black: α-cyanobacterial/non-α-cyanobacterial rubisco). RMSE = 2.1 s$^{-1}$; average explained variance score = 0.55.

