## [Peer Review File · The EMBO Journal]

A systematic exploration of bacterial form I rubisco maximal carboxylation rates

Benoit de Pins, Lior Greenspoon, Yinon Bar-On, Melina Shamshoum, Roei Ben-Nissan, Eliya Milshtein, Dan Davidi, Itai Sharon, Oliver Mueller-Cajar, Elad Noor, and Ron Milo

Corresponding author(s): Ron Milo (Ron.Milo@weizmann.ac.il)

Review Timeline:

Transfer from Review Commons:	1st Feb 24
Editorial Decision:	11th Mar 24
Revision Received:	26th Mar 24
Accepted:	22nd Apr 24

Review
COMMONS

Editor: William Teale

Transaction Report: This manuscript was transferred to The EMBO JOURNAL following peer review at Review Commons.

Review #1

1. Evidence, reproducibility and clarity:

Evidence, reproducibility and clarity (Required)

This is an excellent paper experimentally exploring variations in carboxylation rates of form I rubisco.

I have three comments

1. Q. Did the authors also measure the oxygenate activity of the enzyme? This is relevant to the evolution of carboxysomes and CCMs in general.
2. How do predicted structures (e.g., using Alpha fold) vary with catalytic efficient?
3. The authors should note the paper by Tortell (not this reviewer)
<https://aslopubs.onlinelibrary.wiley.com/doi/pdfdirect/10.4319/lo.2000.45.3.0744>

2. Significance:

Significance (Required)

This is an excellent paper experimentally exploring variations in carboxylation rates of form I rubisco.

3. How much time do you estimate the authors will need to complete the suggested revisions:

Estimated time to Complete Revisions (Required)

(Decision Recommendation)

Cannot tell / Not applicable

4. *Review Commons* values the work of reviewers and encourages them to get credit for their work. Select 'Yes' below to register your reviewing activity at Web of Science Reviewer Recognition Service (formerly Publons); note that

the content of your review will not be visible on Web of Science.

Yes

Review #2

1. Evidence, reproducibility and clarity:

Evidence, reproducibility and clarity (Required)

- I very much enjoyed reviewing this manuscript. de Pins et al provide a timely report on the catalytic turnover rate of a large number of Rubisco enzymes within the FormI group. These data provide novel insights into generalities of Rubisco function, specifically within certain phylogenies, and extend our understanding of carbon acquisition in these systems. In particular, the data presented by de Pins et al provide new insights into the relative carbon fixation rates of alpha-cyanobacteria, for which there are very few studies reporting catalytic turnover. It is apparent that the CO₂ concentrating mechanisms (CCM) of cyanobacteria, especially the alpha-cyanobacteria (containing FormIA Rubisco) are a globally important contributor to CO₂ capture into the biosphere via their carboxysomal Rubisco enzymes. This report provides then first broad selection of FormI Rubiscos to enable comparisons of catalytic turnover across this dominant enzyme family and shows that FormIA Rubiscos from phototrophic systems, and encapsulated in carboxysomes, are on average the fastest enzymes.

- de Pins et al use a high throughput screening technique that provides a highly correlative estimate of Rubisco turnover compared with traditional assays. This screen is based upon bulk expression of enzymes within *E. coli* from synthesised genes, and in some cases the co-expression of chaperonin factors to boost expression and solubility of holoenzymes. The assay process is sound and of high quality and the interpretations clear and uncomplicated.

- The conclusions are sound and I only have a number of minor issues for consideration.

****Minor comments:****

- Temperature effects on 'true' Rubisco turnover rates. The authors quite reasonably note that a single measurement temperature was used in the assay and that this may not necessarily reflect the catalytic turnover of Rubiscos from thermophiles. Suppl. Fig 5b indicates a relatively large number of 'hot spring' species that have, generally, a low median k_{cat} compared with, for example, both cyanobacterial classes. Can the authors comment on whether or not the thermophile set is not highly represented by one group (e.g. phototrophic alpha-cyanobacteria). Does this thermophile dataset have the potential to influence the generalities presented? Fig 2 would suggest this is not the case but it is not possible for the reader to know if all or any thermophiles are represented in Fig2 (as opposed to Suppl. Fig 4).

- Line 98: "in spite of" should be "despite"

- Lines 171-173: There is an additional alpha carboxysomal Rubisco for which there are catalytic parameters described (Chapter 11 Engineering Photosynthetic CO₂ Assimilation to Develop New Crop Varieties to Cope with Future Climates. RE Sharwood, BM Long - Photosynthesis, respiration, and climate change, 2021). This book chapter reports a k_{cat} of 11.9 s⁻¹ for the alpha carboxysomal Rubisco from *Synechococcus* WH8102, very much in line with the authors conclusions.

- Lines 232-234: I note that ref 30 posits that low CO₂ was the more likely driver of carboxysome evolution than high O₂.

- Line 235: The preferred term is either "CO₂ concentrating mechanism" or "inorganic carbon concentrating mechanism"

- Lines 254-256: The relative saturation of carboxysomes with Rubisco is still somewhat undecided, although relatively new datasets enable more accurate comparisons. A number of papers from the Liu Lab (Liverpool) enable estimates of Rubisco active site concentrations for alpha and beta carboxysomes in the range of 2-6 mM. It appears at this stage that Rubisco active site concentrations may be highest in alpha-carboxysomes.

- Lines 312-318: That genes were codon optimized for *E. coli* expression raises an interesting question about the effect of Raf1 on Rubisco solubility. Assuming expression rates were not constrained, can any conclusions be made as to the amino acid sequence differences that led to lower solubility? One assumes that the Rubisco sequences had a high degree of identity?

- Lines 333-344: Was there an attempt to use acRAF (Raf2?) for FormIA Rubiscos that did not fold successfully in E. coli?

2. Significance:

Significance (Required)

This manuscript presents a significant advance in our broader understanding of the major enzyme involved in carbon input into the biosphere, Rubisco. It will be of key interest to those studying carbon biogeochemistry, global CO₂ modelling, cyanobacterial and proteobacterial CCMs, and those interested in using these systems to improve plant-based carbon capture for food security and global carbon abatement systems. It provides, for the first time, a large dataset of hitherto unknown Rubisco kinetics in a globally important group of organisms. The study is extremely well carried out and will likely form the basis of future Rubisco screens to provide greater clarity to our knowledge base of this globally important enzyme.

My expertise is in the study and application of CCMs as CO₂ acquisition systems that can be used for Synbio applications.

3. How much time do you estimate the authors will need to complete the suggested revisions:

Estimated time to Complete Revisions (Required)

(Decision Recommendation)

Less than 1 month

Yes

Review #3

1. Evidence, reproducibility and clarity:

Evidence, reproducibility and clarity (Required)

Summary:

The authors have presented a tremendous study into the diversity of carboxylation speed (k_{cat}) from bacterial Form I Rubisco enzymes. The authors identified some nice diversity in k_{cat} which resulted in the finding that Rubisco's originating from within a CCM were faster, which confirms what has been previously observed in the literature. The authors provided information on a pipeline to screen large numbers of Rubisco variants. In this manuscript, the authors tested 144 different enzymes with 112 of these successfully expressed in E.coli and of these 98 showed substantial catalytic activity. The authors showed that alpha cyanobacterial Rubisco possessed the fastest k_{cat} when compared to beta cyanobacterial counterparts which is contrary to that published in the literature so far. The authors have provided some nice insight into how they improved expression of soluble Rubisco with expressing bacterial chaperonin and Rubisco assembly factors such as Raf1 and rbcX. All which have been previously discovered plant and cyanobacteria. The authors also presented some nice correlations as shown in figure 2 and some weaker and non-correlations to various environmental parameters in the supplementary data.

Overall, the field will learn something from this large body of work that has characterized only one Rubisco catalytic parameter.

Major points:

1) The authors only measured carboxylation speed using a spec assay. The Michaelis constant for CO₂ measured in N₂ and 21% Oxygen is also valuable to understand the diversity in Rubisco catalysis. The authors should perhaps mention this and that the carboxylation efficiency is also an important measure for comparing Rubisco enzymes.

2) The authors mentioned that they used E.coli lysates. Did the authors test for background activity due NADH dehydrogenases which are present in bacterial lysates? This could impact the catalytic rates measured.

3) For the microtitre plate assay, did the authors correct for the different pathlength? This is crucial for the Beer-Lambert law which is used to calculate the consumption of

NADH.

4) Did the authors consider studying the temperature response of kcatc for these enzymes? This could also reveal some interesting insight into their data.

5) With this new catalytic knowledge, what can the field now do with this data to inform new research directions?

****Minor comments:****

The figures are of outstanding quality and easy to follow. This will set the bar high in the literature. I have no other minor comments.

2. Significance:

Significance (Required)

Overall, the authors have presented an excellent study into bacterial Form I Rubisco's that will further enhance our understanding of Rubisco evolution. The pipeline for expression of bacterial Rubisco's in E.coli is developed nicely by the authors and the next step will be to determine how other important catalytic parameters can be determined to have more detailed understanding of Rubisco catalysis.

3. How much time do you estimate the authors will need to complete the suggested revisions:

Estimated time to Complete Revisions (Required)

(Decision Recommendation)

Less than 1 month

No

Full Revision

Manuscript number: RC-2023-02237

Corresponding author(s): Ron, MILO

[Please use this template only if the submitted manuscript should be considered by the affiliate journal as a full revision in response to the points raised by the reviewers.]

*If you wish to submit a preliminary revision with a revision plan, please use our "Revision Plan" template. **It is important to use the appropriate template to clearly inform the editors of your intentions.**]*

1. General Statements [optional]

We thank the reviewers for their time and thought invested in reading the manuscript and providing their feedback, which allows us to greatly improve our manuscript. A point by point response is detailed below, showing how we addressed all the reviewers comments and requests. I hope that the editors and reviewers will consider the updated version suitable for publication.

Thank you for your kind consideration.

**Yours sincerely,
Ron Milo**

This section is mandatory. Please insert a point-by-point reply describing the revisions that were already carried out and included in the transferred manuscript.

Reviewer #1 (Evidence, reproducibility and clarity (Required)):

This is an excellent paper experimentally exploring variations in carboxylation rates of form I rubisco.

We thank the Reviewer for this positive feedback on our study.

I have three comments

Q. Did the authors also measure the oxygenate activity of the enzyme? This is relevant to the evolution of carboxysomes and CCMs in general.

We agree that this is relevant but there are technical limitations for achieving this with the current framework. This pipeline's emphasis on the carboxylation rate allows for the screening of a high number of rubisco variants covering a wide genetic diversity. It provides a way to approach the complexity of the kinetic constraints of this enzyme with a realizable and reproducible method. However, we agree that the measurements of oxygenase activity, as well as affinities to CO₂ and O₂, would enrich our understanding of the evolution of rubiscos in the context of CCMs, and thus expanding our pipeline in the future to cover the other kinetic dimension would be worthwhile but cannot be achieved currently due to various technical constraints (other dimensions to explore, like the temperature effect, could also be included). In line also with the comment of reviewer #3, we have expanded our discussion in the manuscript to clarify this matter more thoroughly:

“By centering our analysis on the carboxylation rate, this pipeline systematically shows the particularity of carboxysome-associated rubiscos which are characterized by a poor affinity to CO₂ (Badger et al, 1998; Falkowski & Raven, 2007) alongside a relatively high carboxylation rate (our data). This likely reflects the aforementioned catalytic tradeoff, suggesting that higher local concentrations of CO₂ within CCMs probably allowed rubiscos to evolve towards higher k_{cat} and K_M . High-throughput measurements of other kinetic parameters beyond what was achieved here, such as the K_M for both gasses or the oxygenation rate, would be valuable. It could provide values of the carboxylation efficiency, or even the enzyme specificity, which would enrich our understanding of this enzyme and of its adaptation to the atmospheric composition over geological timescales.”

2. How do predicted structures (e.g., using Alpha fold) vary with catalytic efficient?

We generated Alpha Fold structures of the 98 active variants revealed in this study and performed preliminary structural analysis of the active site. There was no strong correlation between measured rates and the active site structure. The RMSD of generated structures has a median RMSD of 2.7 Å for the large and small subunit together, and 1.3 Å for the active site, suggesting high conservation of the structure of rubisco, and probably explaining the difficulty to associate rate variation with any clear structural feature (especially coming from prediction algorithm already showing uncertainties on the order of magnitude of an angstrom (Jumper et al, 2021; Terwilliger et al, 2024)). We added a paragraph about this new analysis in the Results and in the Materials and Methods sections of the manuscript, and we present the results in new Supplementary Fig. 12. We made these structures available on the gitlab folder associated with this paper.

Full Revision

3. The authors should note the paper by Tortell (not this reviewer)

<https://aslopubs.onlinelibrary.wiley.com/doi/pdfdirect/10.4319/lo.2000.45.3.0744>

Thank you. We added a reference to this paper in the following sentence: “Another strategy consists of the evolution of rubisco towards stronger affinity for CO₂ (Tortell 2000).”

Reviewer #1 (Significance (Required)):

This is an excellent paper experimentally exploring variations in carboxylation rates of form I rubisco.

Reviewer #2 (Evidence, reproducibility and clarity (Required)):

I very much enjoyed reviewing this manuscript. de Pins et al provide a timely report on the catalytic turnover rate of a large number of Rubisco enzymes within the FormI group. These data provide novel insights into generalities of Rubisco function, specifically within certain phylogenies, and extend our understanding of carbon acquisition in these systems. In particular, the data presented by de Pins et al provide new insights into the relative carbon fixation rates of alpha-cyanobacteria, for which there are very few studies reporting catalytic turnover. It is apparent that the CO₂ concentrating mechanisms (CCM) of cyanobacteria, especially the alpha-cyanobacteria (containing FormIA Rubisco) are a globally important contributor to CO₂ capture into the biosphere via their carboxysomal Rubisco enzymes. This report provides then first broad selection of FormI Rubiscos to enable comparisons of catalytic turnover across this dominant enzyme family and shows that FormIA Rubiscos from phototrophic systems, and encapsulated in carboxysomes, are on average the fastest enzymes.

We thank the Reviewer for the positive remark on the novelty of the study.

de Pins et al use a high throughput screening technique that provides a highly correlative estimate of Rubisco turnover compared with traditional assays. This screen is based upon bulk expression of enzymes within *E. coli* from synthesised genes, and in some cases the co-expression of chaperonin factors to boost expression and solubility of holoenzymes. The assay process is sound and of high quality and the interpretations clear and uncomplicated.

The conclusions are sound and I only have a number of minor issues for consideration.

Minor comments:

Temperature effects on 'true' Rubisco turnover rates. The authors quite reasonably note that a single measurement temperature was used in the assay and that this may not necessarily reflect the catalytic turnover of Rubiscos from thermophiles. Suppl. Fig 5b indicates a relatively large number of 'hot spring' species that have, generally, a low median *k_{cat}* compared with, for

Full Revision

example, both cyanobacterial classes. Can the authors comment on whether or not the thermophile set is not highly represented by one group (e.g. phototrophic alpha-cyanobacteria). Does this thermophile dataset have the potential to influence the generalities presented? Fig 2 would suggest this is not the case but it is not possible for the reader to know if all or any thermophiles are represented in Fig2 (as opposed to Suppl. Fig 4).

We indeed identify 3 groups of rubiscos that are either expressed by thermophilic bacteria (Supplementary Figure 10), and/or are associated with hot environments (hot spring and hydrothermal vent; see Supplementary Figure 5). These rubiscos show relatively lower rates. We grouped them together and reproduced our main analysis (from Figure 2) with and without rubiscos from this group to create a new Figure (Supplementary Fig. 11). Interestingly, alpha-cyanobacteria had no representatives among this group of thermophilic and hot-environments-associated rubiscos. However, this is unlikely to explain the result observed as removing them does not influence the obtained tendencies. We add the following sentences in the Results section:

“Additionally, the slightly lower carboxylation rate of rubiscos originating from thermophilic bacteria and isolated from hot environments (Supplementary Fig. 5 and 10) aligns with expectations, considering that these rubiscos naturally work at higher temperatures than in our *in vitro* assay (30°C). However, this is unlikely to explain the observed trends as the main results of this study are not affected by the removal of these rubiscos (Supplementary Fig. 11).”

We also noticed, while reviewing the study, that the code generating Supplementary Fig. 8 and 10 incorrectly duplicated some dots for a few rubiscos (less than 5), although this did not affect the overall results. We have fixed this issue and have now updated the figures accordingly.

Line 98: "in spite of" should be "despite"

Done.

Lines 171-173: There is an additional alpha carboxysomal Rubisco for which there are catalytic parameters described (Chapter 11 Engineering Photosynthetic CO₂ Assimilation to Develop New Crop Varieties to Cope with Future Climates. RE Sharwood, BM Long - Photosynthesis, respiration, and climate change, 2021). This book chapter reports a *k_{cat}* of 11.9 s⁻¹ for the alpha carboxysomal Rubisco from *Synechococcus* WH8102, very much in line with the authors conclusions.

We added this rubisco to the list of previously characterized rubisco and updated Figure 1 accordingly. We also updated the following sentence:

“However, such statements were made based on scarce measurements, with only three $k_{cat,C}$ values currently available for both rubisco groups (Shih et al, 2016; Long et al, 2018; Sharwood & Long, 2021; Wilson et al, 2018; Aguiló-Nicolau et al, 2023).”

Lines 232-234: I note that ref 30 posits that low CO₂ was the more likely driver of carboxysome evolution than high O₂.

We agree that our phrasing did not point clearly enough that CO₂ was more likely the driver of carboxysome evolution, rather than high O₂. We rephrase the sentence to: “Carboxysomes likely evolved during the Proterozoic eon - in the context of the continuous decrease of carbon dioxide in Earth’s atmosphere (Flamholz & Shih, 2020)”

Line 235: The preferred term is either "CO₂ concentrating mechanism" or "inorganic carbon concentrating mechanism"

We rephrased into “CO₂ concentrating mechanisms”.

Lines 254-256: The relative saturation of carboxysomes with Rubisco is still somewhat undecided, although relatively new datasets enable more accurate comparisons. A number of papers from the Liu Lab (Liverpool) enable estimates of Rubisco active site concentrations for alpha and beta carboxysomes in the range of 2-6 mM. It appears at this stage that Rubisco active site concentrations may be highest in alpha-carboxysomes.

We thank the reviewer for drawing our attention to the work of the Liu lab which, while acknowledging potential inaccuracies in the estimates, tends to show a higher concentration of rubiscos in alpha-carboxysomes than in beta-carboxysomes (Sun et al, 2019; Sun et al, 2022). We removed this statement from the text.

Lines 312-318: That genes were codon optimized for E. coli expression raises an interesting question about the effect of Raf1 on Rubisco solubility. Assuming expression rates were not constrained, can any conclusions be made as to the amino acid sequence differences that led to lower solubility? One assumes that the Rubisco sequences had a high degree of identity?

We indeed note that the fact that every rubisco gene from this study was codon optimized for E. coli expression suggests that the solubility issues met here were post-translational. This suggests a post-translational role for Raf1 in rubisco folding/assembly that is in line with the mechanism proposed by Xia et al. of an interaction of Raf1 with rubisco large subunits dimer, further mediating the assembly of an octameric core and the recruitment of rubisco small subunits (Xia et al, 2020). We also note that insoluble rubiscos were met in all form I clades, suggesting that this property was not linked to a specific group of rubiscos sequences with a high identity degree.

Full Revision

We add that we also tested the effect of not performing codon optimization on 8 rubiscos that were insoluble following codon optimization (to assess whether the codon optimization could negatively affect the folding, for instance by making the translation too fast). This did not yield any improvement in solubility.

Lines 333-344: Was there an attempt to use acRAF (Raf2?) for FormIA Rubiscos that did not fold successfully in *E. coli*?

While only 33% of β -carboxysome-associated (IB) rubiscos were originally soluble (i. e. without the coexpression of Raf1 from *E. natronophila*), 85% of the tested α -carboxysome-associated (IAc) rubiscos were soluble in our experimental conditions. We therefore decided that testing for the effect of co-expressing them with the chaperone acRAF would be less cost-effective.

Reviewer #2 (Significance (Required)):

This manuscript presents a significant advance in our broader understanding of the major enzyme involved in carbon input into the biosphere, Rubisco. It will be of key interest to those studying carbon biogeochemistry, global CO₂ modelling, cyanobacterial and proteobacterial CCMs, and those interested in using these systems to improve plant-based carbon capture for food security and global carbon abatement systems. It provides, for the first time, a large dataset of hitherto unknown Rubisco kinetics in a globally important group of organisms. The study is extremely well carried out and will likely form the basis of future Rubisco screens to provide greater clarity to our knowledge base of this globally important enzyme.

My expertise is in the study and application of CCMs as CO₂ acquisition systems that can be used for Synbio applications.

Reviewer #3 (Evidence, reproducibility and clarity (Required)):

Summary:

The authors have presented a tremendous study into the diversity of carboxylation speed (k_{catc}) from bacterial Form I Rubisco enzymes. The authors identified some nice diversity in k_{catc} which resulted in the finding that Rubisco's originating from within a CCM were faster, which confirms what has been previously observed in the literature. The authors provided information on a pipeline to screen large numbers of Rubisco variants. In this manuscript, the authors tested 144 different enzymes with 112 of these successfully expressed in *E. coli* and of these 98 showed substantial catalytic activity. The authors showed that alpha cyanobacterial Rubisco possessed the fastest k_{catc} when compared to beta cyanobacterial counterparts which is contrary to that published in the literature so far. The authors have provided some nice insight

into how they improved expression of soluble Rubisco with expressing bacterial chaperonin and Rubisco assembly factors such as Raf1 and rbcX. All which have been previously discovered plant and cyanobacteria. The authors also presented some nice correlations as shown in figure 2 and some weaker and non-correlations to various environmental parameters in the supplementary data.

Overall, the field will learn something from this large body of work that has characterized only one Rubisco catalytic parameter.

We thank the Reviewer for these positive comments.

Major points:

1) The authors only measured carboxylation speed using a spec assay. The Michaelis constant for CO₂ measured in N₂ and 21% Oxygen is also valuable to understand the diversity in Rubisco catalysis. The authors should perhaps mention this and that the carboxylation efficiency is also an important measure for comparing Rubisco enzymes.

In line with this and with the comment of reviewer #1, we have expanded the Discussion to include the following:

“By centering our analysis on the carboxylation rate, this pipeline systematically shows the particularity of carboxysome-associated rubiscos which are characterized by a poor affinity to CO₂ (Badger et al, 1998; Falkowski & Raven, 2007) alongside a relatively high carboxylation rate (our data). This likely reflects the aforementioned catalytic tradeoff, suggesting that higher local concentrations of CO₂ within CCMs probably allowed rubiscos to evolve towards higher k_{cat} and K_M . High-throughput measurements of other kinetic parameters beyond what was achieved here, such as the K_M for both gasses or the oxygenation rate, would be valuable. It could provide values of the carboxylation efficiency, or even the enzyme specificity, which would enrich our understanding of this enzyme and of its adaptation to the atmospheric composition over geological timescales.”

2) The authors mentioned that they used E.coli lysates. Did the authors test for background activity due NADH dehydrogenases which are present in bacterial lysates? This could impact the catalytic rates measured.

The background activity due to dehydrogenases from the lysate is negligible compared to the measured reaction (see as an example, in Supplementary Fig. 16A, the gray curve, corresponding to a CABP concentration of 90 nM, fully inhibiting rubisco in the reaction). Moreover, the use of CABP in the spectroscopic assay allows to take into account any “background activity” that would come from an element independent of rubisco because this background would be identical at every CABP concentration. We modified Supplementary Note 1 to make this point clearer:

“We note that any NADH dehydrogenation due to other native *E. coli* proteins, while low (see the [CABP] = 90 nM gray curve of Supplementary Fig. 16A), is not influencing the measurement of the carboxylation rate which relies on the differential V_{\max} values at changing CABP concentrations (which should not affect the rates of these dehydrogenases).”

3) For the microtitre plate assay, did the authors correct for the different pathlength? This is crucial for the Beer-Lambert law which is used to calculate the consumption of NADH.

Because our assay is performed in a microwell plate instead of a standard 1 cm quartz cuvette, we indeed calibrated the assay to empirically determine the optical path length in our conditions. We add the following sentence in the Material and Methods section to better explain the measurement of NADH concentration in our assay:

“Knowing the NADH extinction coefficient at 340 nm ($\epsilon_{340} = 6220 \text{ M}^{-1}\text{cm}^{-1}$), and after measuring the optical path length ($l = 0.26 \text{ cm}$) with an NADH calibration curve in our setting, we used Beer-Lambert law ($A_{340} = \epsilon_{340} \cdot l \cdot c$) to measure the NADH concentration c .”

4) Did the authors consider studying the temperature response of k_{cat} for these enzymes? This could also reveal some interesting insight into their data.

We indeed considered this. With a high-throughput pipeline involving >100 enzymes to express and test in parallel, we had to limit the experimental testing conditions to be realistic both in terms of time and budget. However, the finding that rubisco variants from thermophiles tend to have lower carboxylation rates in our standardized conditions (30°C) suggest that they will probably show faster rates at temperatures closer to their optimal growth temperature. We therefore agree that the study of the temperature response of the carboxylation rate in further works could validate these hypotheses and bring more insight into the catalytic characteristics of rubisco. We further emphasize this point in the following modified sentence:

“Investigating the temperature response of rubisco carboxylation rate in further work could shed light on the importance of this parameter, especially among thermophilic or psychrophilic associated enzymes.”

5) With this new catalytic knowledge, what can the field now do with this data to inform new research directions?

We believe this new knowledge can inform new research directions in the field of microbial ecology, metabolic engineering, and machine learning in the context of kinetic parameters prediction. We modified a paragraph in the discussion to elaborate on this point:

Full Revision

“This study provides a systematic exploration of bacterial form I rubisco maximal rates and its relationship with various contextual factors that could have shaped the evolution of this most abundant enzyme on Earth. It holds potential for future metabolic and ecological studies about specific bacterial species – for instance among cyanobacteria for which 40 rubiscos have been characterized here. By enriching our knowledge on carboxylation rates and their connection to environmental factors, it can also contribute to more accurately modeling global carbon fluxes. Additionally, this dataset of rubisco sequences and their associated rates can facilitate linking sequence motifs to catalytic function. Ultimately, it can improve our understanding, and possible harnessing, of bacterial CCMs for the potential development of plant-based carbon capture strategies, the increase of agricultural yields and the support of sustainable food production in the face of a changing climate.”

Minor comments:

The figures are of outstanding quality and easy to follow. This will set the bar high in the literature. I have no other minor comments.

We thank the Reviewer for noting the care taken in the graphic presentation of our results.

Reviewer #3 (Significance (Required)):

Overall, the authors have presented an excellent study into bacterial Form I Rubisco's that will further enhance our understanding of Rubisco evolution. The pipeline for expression of bacterial Rubisco's in E.coli is developed nicely by the authors and the next step will be to determine how other important catalytic parameters can be determined to have more detailed understanding of Rubisco catalysis.

Dear Prof. Milo,

Thank you for submitting your manuscript for consideration by the EMBO Journal. It has now been seen by three referees whose comments are enclosed. As you will see, all three referees express interest in your manuscript and are in favour of publication, pending satisfactory minor revision.

We are currently running our editorial checks and will get back to you soon with any minor formatting issues that may arise. My colleague Hannah Sonntag will contact you separately about your providing the necessary source data to accompany the final article.

Thank you for the opportunity to consider your work for publication. I look forward to moving forward with this work.

Yours sincerely,

William Teale

William Teale, PhD
Editor
The EMBO Journal
w.teale@embojournal.org

We realize that it is difficult to revise to a specific deadline. In the interest of protecting the conceptual advance provided by the work, we recommend a revision within 3 months (9th Jun 2024). Please discuss the revision progress ahead of this time with the editor if you require more time to complete the revisions. Use the link below to submit your revision:

Referee #1:

I am completely satisfied with the revisions provided in this version of the authors' manuscript and am eager to see it published in its current form. I congratulate the authors on a superb piece of work.

Referee #2:

The authors have addressed the major concerns I had.
It should be published - and, like most papers, it is a conversation in science.

Referee #3:

The authors have investigated the diversity of Rubisco carboxylation speed (k_{cat}) among bacterial Form I Rubisco's. This is an exceptional study where the authors attempted to express in E.coli 144 Rubisco operons with 112 of them being soluble and of these 98 were able to be catalytically tested. As expected, the authors showed that CCM's had faster Rubisco enzymes and that alpha-cyanobacterial Rubisco had a faster Rubisco compared to the beta-cyanobacterial counterparts. The authors showed that in addition to expressing GroEL and GroES, Raf1 was required to enhance soluble recovery of recombinant Rubisco. It is important to point out that this study was completed at 30C which is different from the majority of Rubisco parameters published in the literature. Therefore, comparisons to Rubisco's measured at 25C can be difficult. Nevertheless, this comparative study was able to draw some nice conclusions.

Overall, this is a high quality dataset that will be very informative to the field. It will augment further studies into the diversity of Rubisco catalytic parameters among the Rubisco superfamily.

A couple of minor comments:

Supplementary Figure 15 - Did the authors load this gel on a protein basis? Can you authors provide more detail? ie how many ug per lane etc.

Lines 251 - 252 - The authors should qualify this statement by outlining this is the most comprehensive comparative study of Rubisco catalytic speed presented to date.

Referee #1:

I am completely satisfied with the revisions provided in this version of the authors' manuscript and am eager to see it published in its current form. I congratulate the authors on a superb piece of work.

We deeply thank the referee for the insightful comments which allowed us to greatly improve our manuscript.

Referee #2:

The authors have addressed the major concerns I had.
It should be published - and, like most papers, it is a conversation in science.

We fully agree. We thank the referee for the supportive feedback and for taking part in this fruitful conversation.

Referee #3:

The authors have investigated the diversity of Rubisco carboxylation speed (k_{cat}) among bacterial Form I Rubisco's. This is an exceptional study where the authors attempted to express in E.coli 144 Rubisco operons with 112 of them being soluble and of these 98 were able to be catalytically tested. As expected, the authors showed that CCM's had faster Rubisco enzymes and that alpha-cyanobacterial Rubisco had a faster Rubisco compared to the beta-cyanobacterial counterparts. The authors showed that in addition to expressing GroEL and GroES, Raf1 was required to enhance soluble recovery of recombinant Rubisco. It is important to point out that this study was completed at 30C which is different from the majority of Rubisco parameters published in the literature. Therefore, comparisons to Rubisco's measured at 25C can be difficult. Nevertheless, this comparative study was able to draw some nice conclusions. Overall, this is a high quality dataset that will be very informative to the field. It will augment further studies into the diversity of Rubisco catalytic parameters among the Rubisco superfamily.

We thank the referee for this positive conclusion. We believe their review has enhanced the quality of our manuscript and we are deeply appreciative to them.

A couple of minor comments:

Supplementary Figure 15 - Did the authors load this gel on a protein basis? Can you authors provide more detail? ie how many ug per lane etc.

More detail was provided on how the SDS-Page were loaded in the modified sentence:

“For quality control of each sample, 0.2 µl of the crude extracts and 2 µl of the soluble fractions (i.e., 20-40 ng of proteins), were run on an SDS-PAGE gel (Appendix Fig. S12).”

Lines 251 - 252 - The authors should qualify this statement by outlining this is the most comprehensive comparative study of Rubisco catalytic speed presented to date.

We corrected the sentence accordingly:

“Our findings with carboxysomal rubiscos support this conjecture using the most comprehensive comparative study of rubisco catalytic speed presented to date.”

Dear Prof. Milo,

I am pleased to inform you that your manuscript has been accepted for publication in the EMBO Journal.

Congratulations on a very insightful and interesting study!

Yours sincerely,

William Teale

William Teale, PhD
Editor
The EMBO Journal
w.teale@embojournal.org
